# MoT: Memory-of-Thought Enables ChatGPT to Self-Improve

## Xiaonan Li, Xipeng Qiu

School of Computer Science, Fudan University
Shanghai Key Laboratory of Intelligent Information Processing, Fudan University
{lixn20, xpqiu}@fudan.edu.cn

## Abstract

Large Language Models (LLMs) have shown impressive abilities in various tasks. However, fundamentally improving them depends on high-quality datasets or computationally expensive fine-tuning. On the contrary, humans can easily improve themselves by self-thinking and memory, without external resources. In this paper, we propose a framework, **MoT**, to let the LLM self-improve through **M**emory-**o**f-**T**hought, without annotated datasets and parameter updates. Specifically, MoT is divided into two stages: 1. before the test stage, the LLM pre-thinks on the unlabeled dataset and saves the high-confidence thoughts as external memory; 2. During the test stage, given a test question, the LLM recalls relevant memory to help itself reason and answer it. Experimental results show that MoT can help ChatGPT significantly improve its abilities in arithmetic reasoning, commonsense reasoning, factual reasoning, and natural language inference. Further analyses show that each component contributes critically to the improvements and MoT can lead to consistent improvements across various CoT methods and LLMs.

## 1 Introduction

Large Language Models (LLMs) have demonstrated surprising abilities on a wide range of Natural Language Processing (NLP) tasks (Chen et al., 2023; Zhang et al., 2022a; Chowdhery et al., 2022; Tay et al., 2022; OpenAI, 2023; Hoffmann et al., 2022; Touvron et al., 2023; Mialon et al., 2023; Zhao et al., 2023; Qiu et al., 2020). Notably, new abilities emerge in LLMs as they are scaled to hundreds of billions of parameters, like in-context few-shot learning (Chen et al., 2023; Dong et al., 2022), simple digit operation and factual knowledge query (Wei et al., 2022b). Especially, the general reasoning ability of the LLM has impressed the NLP community and relevant techniques have achieved a series of new state-of-the-art (Wei et al., 2022c; Kojima et al., 2022; Lampinen et al., 2022;

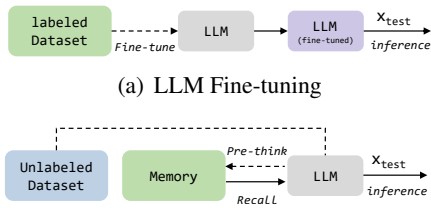

(a) LLM Fine-tuning

(b) Pre-thinking and Recalling

Figure 1: The comparison between fine-tuning and MoT: while fine-tuning LLM with labeled datasets is costly and needs powerful computational resources, MoT can make the LLM self-improve via pre-thinking and recalling, without parameter updates and annotated datasets.

Wang et al., 2022b; Huang and Chang, 2022). Specifically, Wei et al. (2022c) and Kojima et al. (2022) propose few-shot CoT and zero-shot CoT, which elicit LLM's reasoning by few-shot demonstrations and simple yet effective "*Let's think step by step*" prompting, respectively. Based on them, Wang et al. (2022b); Press et al. (2022); Zhou et al. (2022); Wang et al. (2023); Weng et al. (2022) further propose self-consistency, self-ask, least-to-most, plan-and-solve, etc., to achieve more complicated reasoning in various specialized scenarios.

Despite the impressive abilities of the LLM pretrained on the large corpus, fundamentally improving the LLM's performance beyond few-shot / zero-shot baselines highly depends on either high-quality annotated datasets or costly fine-tuning of LLMs. In general, these methods can be divided into three categories: 1. Annotated Datasets + Fine-tuning: Wei et al. (2022a) and Sanh et al. (2022) propose FLAN and T0 respectively to enhance the LLM's zero-shot ability by tens of curated NLP benchmark datasets. Based on FLAN, Chung et al. (2022) scale up its training in terms of model size and the number of tasks, and demonstrate that the added CoT examples with rationales improve the LLM's reasoning abilities. Instruct-GPT (Ouyang et al., 2022) improves the GPT-3's instruction-following ability by fine-tuning on many diverse crowd-sourced instruction-answer

pairs. 2. Retrieving Annotated Data: Liu et al. (2022), Su et al. (2022a) and Agrawal et al. (2022) use SentenceBERT (Reimers and Gurevych, 2019) or BM25 (Robertson and Zaragoza, 2009) to retrieve relevant examples from the annotated dataset, to improve LLM's in-context learning. Rubin et al. (2022) and Shi et al. (2022a) leverage annotated datasets to train retrievers by the LM-feedback to retrieve helpful demonstrations for the test example. 3. Fine-tuning with LLM-generated data: Zelikman et al. (2022) let the LM generate rationales for annotated dataset and train itself to enhance the reasoning ability. Magister et al. (2022),Ho et al. (2022) and Fu et al. (2023a) use the reasoning paths generated by large LM to improve the small LM's reasoning capability. More recently, Huang et al. (2022) demonstrate the effectiveness of self-training on PaLM (Chowdhery et al., 2022).

As annotating high-quality data, especially rationales in CoT data, is expensive, fine-tuning LLM requires extremely powerful computational resources and results in high computational costs. Methods above that rely on fine-tuning also face two challenges: 1. Since the most powerful LLMs, e.g., GPT-4 (OpenAI, 2023) and PaLM (Chowdhery et al., 2022; Anil et al., 2023), are only publicly available through the inference API, it is not feasible for most of the research community to improve them by these methods. 2. Fine-tuning LLM for specific capability enhancement is costly and not environmentally friendly. As the LLM has massive parameters, fine-tuning them will lead to substantial costs of model storage and deployment. Further studies show that fine-tuning the LLM with specialized data may significantly decrease its general abilities (Fu et al., 2023b).

While considerable efforts were dedicated to collecting high-quality annotated datasets and fine-tuning the LLM, which is costly and may decrease its general ability, on the contrary, humans can improve their own reasoning abilities through the metacognition process (Dunlosky and Metcalfe, 2008) and the memory mechanisms (Tulving, 2002), and preserve their general abilities. For example, memory helps humans improve themselves in terms of decision-making, reasoning, judgment, etc (Tulving, 2002). Inspired by this, we propose **MoT**, shown in Figure 1, a pre-think-then-recall framework to let the LLM self-improve through **M**emory-**o**f-**T**houghts, without supervised data and parameter updates. In the pre-thinking stage, the LLM thinks on the unlabeled dataset and saves the

thoughts as external memory. In the test stage, the LLM recalls relevant memory to help reason and answer the given test question. Since we focus on the overall framework and aim to demonstrate its generality and extensibility, we use simple components to instantiate these two stages. Specifically, we use the simple Few-Shot-CoT (Wei et al., 2022c) with multiple-path decoding strategy (Wang et al., 2022b) in the pre-thinking stage and propose answer-entropy to filter out uncertain thoughts. For memory recall, we propose LLM-retrieval, which *lets the LLM itself retrieve* relevant memory to help answer the test question. Compared with typical semantic retrievers like SBERT (Reimers and Gurevych, 2019), LLM-retrieval can better capture the deep connection of complicated logic and reasoning than semantic embeddings.

We summarize our contribution as follows:

- To the best of our knowledge, the proposed framework is the first to let LLM improve its own reasoning abilities based on the memory mechanism, without parameter updates and annotated datasets.

- We conduct comprehensive experiments on extensive datasets and the results show that MoT can help ChatGPT improve its abilities in arithmetic reasoning, commonsense reasoning, factual reasoning and natural language inference without parameter updates and annotated datasets. Further analyses show that each component contributes critically to the improvements and MoT can lead to consistent improvements across various CoT methods and LLMs.

- We release the code and generated CoT reasoning paths to facilitate future research[1]. In this paper, we instantiate the proposed framework with simple components and demonstrate its effectiveness. We hope that MoT can inspire researchers of the important design choices about making the LLM self-improve with memory mechanisms and pave the way for further improvements.

## 2 Background: Chain of Thought

The large language model has shown impressive reasoning abilities on various tasks. Chain-of-Thoughts (CoT) prompting (Wei et al., 2022c; Kojima et al., 2022) is the most prevailing way to

---

[1]https://github.com/LeeSureman/MoT

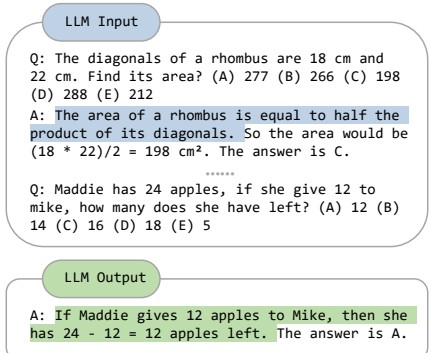


**LLM Input**

Q: The diagonals of a rhombus are 18 cm and 22 cm. Find its area? (A) 277 (B) 266 (C) 198 (D) 288 (E) 212
A: The area of a rhombus is equal to half the product of its diagonals. So the area would be (18 * 22)/2 = 198 cm². The answer is C.

......

Q: Maddie has 24 apples, if she give 12 to mike, how many does she have left? (A) 12 (B) 14 (C) 16 (D) 18 (E) 5

**LLM Output**

A: If Maddie gives 12 apples to Mike, then she has 24 - 12 = 12 apples left. The answer is A.


Figure 2: The illustration of Few-Shot-CoT.

let the LLM reason, i.e., generate a series of intermediate reasoning steps that lead to the final answer. As shown in Figure 2, Few-Shot-CoT (Wei et al., 2022c; Lampinen et al., 2022) provides a few demonstrations with rationales, i.e., question/rationale/answer pairs, and prompts the LLM to generate the rationale that leads to the final answer. Zero-Shot-CoT (Kojima et al., 2022) adds the prompt, "*Let's think step by step*", after the test question and elicits the LLM's reasoning. Specifically, the Few-Shot-CoT gets the answer as:

$$s = \text{LLM}(d_1, d_2, \cdots, q_{test}) \tag{1}$$

$$a = \text{Parse-Answer}(s), \tag{2}$$

where $d_i = [x_i, r_i, a_i]$ is the $i$ th demonstration and consists of the input, rationale and answer. Few-Shot-CoT first decodes $s$ from the LLM given the few-shot CoT demonstrations, and parses $s$ to get the final answer. Since the demonstration is typically in the format: "[input] [rationale] *The answer is* [answer]", the answer can be easily parsed from $s$ by the trigger "*The answer is*" (Wei et al., 2022c). Similarly, Zero-Shot-CoT uses answer triggers, e.g., "*Therefore, the answer is*", to extract the final answer from the zero-shot reasoning path generated by LLM (Kojima et al., 2022).

## 3 Method

We show the overview of our framework in Figure 3. In this paper, we mainly focus on making LLM self-improve in the typical few-shot CoT scenario, where we are given a frozen large language model and an unlabeled dataset with a few CoT demonstrations (Wei et al., 2022c; Huang et al., 2022). We further demonstrate MoT's effectiveness in zero-shot scenarios in section 4.3. Our framework is divided into two stages: **1. Pre-Think** Before the test stage, the LLM thinks over the unlabeled dataset and keeps the high-confidence reasoning paths as memory. **2. Recall** In the test stage,

given a test question, we propose LLM-retrieval to let the *LLM retrieve relevant memory to help itself* reason and answer it. Our method does not depend on high-quality labeled datasets and costly fine-tuning of LLM, and it is feasible when the LLM is frozen or only available through the inference API. Since we let the LLM think over the unlabeled dataset, save the self-generated thoughts as external memory and retrieve relevant memory for itself to help reasoning, we consider our method as making the LLM self-improve with *Memory-of-Thought*. We introduce these two stages below.

### 3.1 Pre-Thinking

#### 3.1.1 Let LLM Think before Test Stage

In this stage, we let the LLM think over the unlabeled dataset and save the resultant question/rationale/answer pairs as external memory. Since we focus on the overall framework and aim to demonstrate its generality and extensibility, we instantiate the "thinking" mechanism here as the simple Few-Shot-CoT (Wei et al., 2022c) with multiple-path decoding strategy (Wang et al., 2022b) in this paper. Specifically, for each example $x$ from the unlabeled dataset $X$, we let the LLM sample $n$ reasoning paths and answers with temperature $T > 0$, denoted as $[r_1, r_2 \cdots, r_n]$ and $[a_1, a_2 \cdots, a_n]$. Then we use majority-voting to select the most consistent answer, $\widetilde{a} = \arg\max_{a_i} \sum_{j=1}^{n} \mathbb{1}(a_i = a_j)$, and keep the reasoning path, which leads to $\widetilde{a}$, as memory. Since we only consider the thought that leads to the most consistent answer, the retained thoughts can be more accurate (Wang et al., 2022b) and better help the test stage. For simplicity and to save memory size, we randomly select one reasoning path of the final answer for each unlabeled example and see saving multiple thoughts for one question as future work.

#### 3.1.2 High-Confidence Thought Filtering

Since the most consistent answer does not necessarily lead to the correct answer and incorrect demonstrations can cause inferior performance (Yoo et al., 2022; Lyu et al., 2022), we further propose to filter the thoughts by uncertainty. Inspired by Liu et al. (2020); Xin et al. (2020), we propose the answer-entropy $u(\cdot)$ to filter out high-uncertainty thoughts:

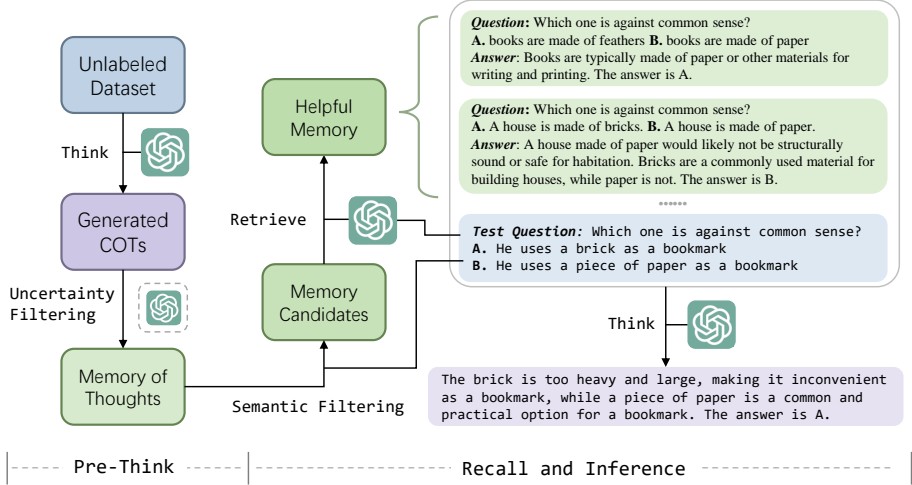

Figure 3: The overview of MoT.

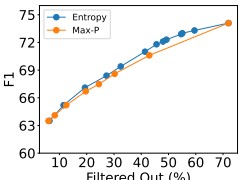

Figure 4: The relation between accuracy and the filter-out ratio, after multiple-paths decoding on the DROP dataset. Under the same filter-out ratio, filtering by answer entropy leads to slightly higher F1-score than filtering by Max-P proposed by Huang et al. (2022).

$$A^* = \text{unique}(\{a_i\}_{i=1}^n) \quad (3)$$

$$p(a_i^*) = \sum_{j=1}^n \mathbb{1}(a_i^* = a_j)/n \quad (4)$$

$$u(a_i^*) = -\sum_{i=1}^{|A^*|} p(a_i^*) \log p(a_i^*) \quad (5)$$

where $A^* = \{a_1^*, a_2^* \cdots\}$ is the set of answers. $u(\cdot)$ indicates the answer uncertainty, and the higher $u(\cdot)$ is, the more uncertain the LLM is. We filter out thoughts whose uncertainty is higher than $\tau$ and $\tau$ is a pre-defined threshold. In the exploratory experiment (Figure 4), we find the thought with lower uncertainty is more likely to be correct: the stricter the filtering is, the more accurate the remaining thoughts are. Hence the answer-entropy can filter out noisy thoughts and lead to more accurate thoughts for recalling. Compared with the filtering in Huang et al. (2022), which uses the number of consistent paths (max probability, abbreviated as Max-P) as metric, answer-entropy leads to slightly higher accuracy, under the same filter-out ratio.

After filtering, we obtain the pool of memory-of-thoughts, $M = \{m_i\}_{i=1}^{|M|}$, where $m_i$ is the concatenation of corresponding input, reasoning path and answer (see Figure 3). $M$ consists of the high-quality thoughts of LLM on various questions and thus contains crucial and valuable information for the LLM to answer the test question. For the coherence in the subsequent content, we will refer to $m_i$ as "memory" or "thought".

## 3.2 Recalling

In the test stage, the relevant memory is retrieved from the memory pool $M$, to help the LLM answer the given test question, $q_{test}$. Although semantic embedders, e.g., SBERT (Reimers and Gurevych, 2019) are capable of retrieving semantically relevant examples for ICL (Liu et al., 2022), for reasoning tasks, it is challenging for them to fully capture the deep logical connections between $q_{test}$ and helpful memory, as a single vector can not directly reflect the intricate logic and reasoning path. Since the LLM, e.g., ChatGPT, has shown impressively powerful and general natural language understanding capability and a certain level of self-awareness (Kadavath et al., 2022), we propose LLM-retrieval to let the LLM retrieve helpful memory for itself.

As the LLM has a limitation of the max length, it is infeasible to let the LLM directly select among the entire memory pool. Inspired by human's memory recall process, where we usually first unconsciously filter the relevant memories and then consciously evaluate them (Schacter and Addis, 2007; Franklin et al., 2005), we divide LLM-retrieval into two stages: 1. filter out semantically irrelevant memory and get memory candidates; 2. let the LLM choose from memory candidates.

Since the diversity of demonstrations has been shown important for LLMs(Ye et al., 2023; Levy et al., 2022; Li and Qiu, 2023), we follow Li et al. (2022) to conduct memory retrieval with diversity-

based clustering, i.e., we partition the entire memory pool into $l$ clusters, $\{M^{(1)}, M^{(2)} \cdots, M^{(l)}\}$, and retrieve one memory from each cluster separately. Specifically, for each cluster $M^{(i)}$, we first use an off-the-shelf semantic embedder, e.g., SBERT (Reimers and Gurevych, 2019), to filter out semantically irrelevant memory and get memory candidates as follows:

$$M_c^{(i)} = \text{top-k}_{m \in M^{(i)}}(\text{sim}(q_{test}, m)), \quad (6)$$

where $\text{sim}(\cdot, \cdot)$ is the cosine similarity of semantic embeddings. $M_c^{(i)}$ are the $i$-th cluster's candidates and contain $k$ memories.

Then we further let the LLM select the most helpful memory from each cluster as follows:

$$m^{(i)} = \text{LLM}(q_{test}, M_c^{(i)}, P_{retrieval}), \quad (7)$$

where $P_{retrieval}$ is the prompt for the LLM to retrieve helpful memory. We concatenate the test question $q_{test}$, memory candidates $M_c^{(i)}$ and $P_{retrieval}$ by a specialized template. The resulting input for LLM is like: "*References: $[M_{c,1}^{(i)}, M_{c,2}^{(i)} \cdots, M_{c,k}^{(i)}]$ Target Question: $[q_{test}]$ which one reference would be the most helpful for you to answer the target question?*".

In this manner, we can utilize the LLM's powerful natural language understanding ability to select the most helpful memory of $M_c^{(i)}$ for itself. Since these retrieved memories are from diverse memory clusters, they can be not only helpful for $q_{test}$ but also comprehensive, thus facilitating the LLM to answer the test question. Meanwhile, the semantical-filtering can filter out semantically irrelevant memories in advance, thus significantly helps save the number of LLM calls.

In exploratory experiments, we find that providing only memory candidates' questions for LLM-retrieval almost does not affect the retrieval result. Hence, for each $M_c^{(i)}$, we only provide its question for the LLM to select, which can significantly save the inference cost of the LLM. To make LLM better understand the goal of retrieving helpful memory and make its output easy-parsing with a pre-defined format, we append extra instructions like "*You must end in the format like "The most helpful question is question [idx]."*" to the input. We show the complete input of LLM-retrieval in Appendix A.

### 3.3 Inference

Given a test question $q_{test}$, the LLM can think and then output the answer based on the retrieved memory, $m^{(1)}, m^{(2)} \cdots m^{(k)}$. Specifically, we let the LLM reason in the manner of Few-Shot-CoT:

$$s = \text{LLM}(m^{(1)}, m^{(2)}, \cdots, m^{(k)}, q_{test}) \quad (8)$$

$$a = \text{Parse-Answer}(s) \quad (9)$$

In this paper, we focus on the overall framework and instantiate it with simple components, i.e., Few-Shot-CoT, simple uncertainty filtering and LLM-retrieval. We further analyze the orthogonality of MoT and different CoT methods in section 4.3. We leave exploring more implementations, e.g., letting the LLM itself filter out uncertain thoughts (Weng et al., 2022; Long, 2023), as the future work.

## 4 Experiment

### 4.1 Experimental Settings

**Dataset** We conduct experiments on ten datasets, across four task families: **Arithmetic reasoning:** AQuA (Ling et al., 2017) and DROP (Dua et al., 2019); **Natural Language Inference:** Adversarial NLI subsets (Nie et al., 2020), including ANLI-A1, ANLI-A2 and ANLI-A3, which cover varying difficulty levels; **Commonsense Reasoning:** OBQA (Mihaylov et al., 2018) and ComV (Commonsense Validation) (Wang et al., 2019); **Factual Reasoning:** BoolQ (Clark et al., 2019), FactCK (Fact Checker) and WikiQA (Srivastava et al., 2022). We list dataset overview, statistics, split and evaluation metrics in Appendix B.

**Method Comparison** Since we focus on whether MoT can help the LLM self-improve, we compare MoT with baselines on the same LLM, ChatGPT (GPT-3.5-Turbo-0301), including zero-shot/few-shot CoT and zero-shot/few-shot direct prompting. To analyze the effect of rationales and thinking in MoT, we additionally compare MoT with its two variants: 1) MoT (no rationale), which removes rationales in the retrieved memory and thus lets the LLM directly output the answer, which can be seen as the few-shot direct version of MoT; 2) MoT (no thinking), which keeps rationales in the retrieved memory but forces the LLM to directly answer the question without CoT. Specifically, we add "*The answer is*" as the LLM's output prefix to prompt the LLM directly output the answer. Through these two variants, we can analyze the effect of rationales and the thinking in MoT, respectively. Additionally, we conduct experiments of MoT under annotated datasets, MoT (with gold), to see its potential improvement space, where we use the gold labels to filter out incorrect memory. Thus, MoT will not be degraded by the incorrect answer.

**Implementation Details** We use the public OpenAI language model of "gpt-3.5-turbo-0301" un-

| Method | Arithmetic Reasoning | | ANLI | | | CS Reasoning | | Factual Reasoning | | | AVG |
|---|---|---|---|---|---|---|---|---|---|---|---|
| | AQuA | DROP | -A1 | -A2 | -A3 | OBQA | ComV | BoolQ | FactCK | WikiQA | |
| Zero-Shot | 27.7 | 24.7 | 54.4 | 48.0 | 51.7 | 79.4 | 90.5 | 63.4 | 75.6 | 52.6 | 56.8 |
| Few-Shot | 28.9 | 46.3 | 55.0 | 48.5 | 51.1 | 82.0 | 90.8 | 64.4 | 77.0 | 32.5 | 57.6 |
| MoT (no rationale) | 27.0 | 59.4 | 56.2 | 50.3 | 52.6 | **84.2** | 91.0 | 70.1 | 82.1 | 53.9 | 62.7 |
| MoT (no thinking) | 24.4 | 59.4 | 55.6 | 50.2 | 52.6 | 81.3 | 90.5 | 71.6 | 82.2 | 64.3 | 63.1 |
| Zero-Shot-CoT | 51.7 | 62.2 | 61.9 | 51.6 | 48.5 | 69.2 | 87.1 | 53.0 | 66.0 | 49.9 | 60.1 |
| Few-Shot-CoT | 49.7 | 57.6 | 59.7 | 48.1 | 52.3 | 80.0 | 94.5 | 67.7 | 80.6 | 65.2 | 65.5 |
| MoT | **54.1** | **65.7** | **64.6** | **52.8** | **55.2** | 82.3 | **95.5** | **71.5** | **82.2** | **68.0** | **69.2** |
| MoT (with gold) | 56.5 | 71.0 | 65.7 | 55.6 | 55.4 | 82.8 | 94.6 | 74.2 | 86.6 | 70.6 | 71.3 |

Table 1: Performance comparison on ChatGPT (GPT-3.5-Turbo-0301).

less otherwise specified and the experiments on "text-davinci-002/003" (Appendix C) show consistent trends. For recalling, we use SBERT ("all-mpnet-base-v2") (Reimers and Gurevych, 2019) for semantic filtering. And we set the number of clusters $l$ (also the demonstration quantity) and the number of each cluster's memory candidates $k$ as 4 and 10, respectively. We further analyze the number of demonstrations in Appendix D. In the test stage, for the stability of results, we use greedy decoding to generate the output, unless otherwise specified. We list the full implementation details and few-shot demonstrations in Appendix E.

### 4.2 Main Results

We show the results in Table 1. We see that MoT significantly outperforms baselines on most datasets, which shows MoT's best comprehensive reasoning capability on a series of NLP tasks. Specifically, MoT exceeds Few-Shot-CoT and Zero-Shot-CoT by 3.7 and 9.1 points respectively, and this directly demonstrates that MoT can make the LLM improve itself by memory-of-thoughts, without annotated dataset and parameter updates. Notably, Zero-Shot-CoT shows impressive performance on ChatGPT and outperforms Few-Shot-CoT on several datasets, e.g., AQuA, DROP, ANLI-1 and ANLI-2, which indicates the potential unnecessity of irrelevant CoT demonstration for the LLM with powerful zero-shot reasoning ability. Meanwhile, MoT surpasses Zero-Shot-CoT consistently on all datasets and this indicates the helpfulness of retrieved memory.

As for MoT's two variants, they also show better overall performance than Zero-Shot and Few-Shot. Meanwhile, despite directly outputting the answer, they outperform Zero-Shot-CoT and Few-Shot-CoT on several datasets, e.g., OBQA, BoolQ and FactCK. This is analogous to a common phenomenon in human beings: when recalling relevant

| Method | OBQA | BoolQ | WikiQA |
|---|---|---|---|
| *Decoding Paths=8* | | | |
| Zero-Shot-CoT$_{T=0.7}$ | 80.4 | 55.9 | 59.2 |
| Zero-Shot-CoT$_{T=1}$ | 82.4 | 54.1 | 59.2 |
| Zero-Shot-CoT$_{T=1.2}$ | 81.0 | 49.1 | 59.4 |
| Few-Shot-CoT$_{T=0.7}$ | 82.0 | 69.2 | 69.6 |
| Few-Shot-CoT$_{T=1}$ | 83.6 | 68.2 | 70.8 |
| Few-Shot-CoT$_{T=1.2}$ | 82.2 | 70.2 | 70.8 |
| MoT$_{T=0.7}$ | **85.0** | 73.2 | **73.2** |
| MoT$_{T=1}$ | 84.4 | **73.5** | 72.1 |
| MoT$_{T=1.2}$ | 85.0 | 72.8 | 72.1 |
| *Decoding Paths=16* | | | |
| Zero-Shot-CoT$_{T=0.7}$ | 81.6 | 58.0 | 63.8 |
| Zero-Shot-CoT$_{T=1}$ | 83.2 | 55.8 | 65.7 |
| Zero-Shot-CoT$_{T=1.2}$ | 83.4 | 53.8 | 64.3 |
| Few-Shot-CoT$_{T=0.7}$ | 83.2 | 69.6 | 71.5 |
| Few-Shot-CoT$_{T=1}$ | 83.6 | 69.2 | 71.7 |
| Few-Shot-CoT$_{T=1.2}$ | 83.6 | 69.7 | 70.9 |
| MoT$_{T=0.7}$ | 84.4 | **73.7** | 73.7 |
| MoT$_{T=1}$ | **85.0** | 72.7 | **74.2** |
| MoT$_{T=1.2}$ | 84.8 | 73.3 | 74.1 |

Table 2: Performance comparison under self-consistency strategy across various hyper-parameters.

memory, we can perform well by intuition, without conscious reasoning (Dijksterhuis and Nordgren, 2006; Todd et al., 1999). Additionally, although MoT (no thinking) is provided with the rationales while MoT (no rationale) is not, they show generally similar performance, which indicates that explicit reasoning is necessary for the LLM to fully leverage the retrieved memory. In short, both relevant memory and explicit reasoning are essential for MoT to consistently achieve improvements on extensive datasets.

Additionally, MoT (with gold) shows better performance than MoT, which indicates the potential improvements when MoT applies more advanced CoT methods (Wang et al., 2023; Zhou et al., 2022; Zheng et al., 2023; Long, 2023) and verification methods (Weng et al., 2022; Manakul et al., 2023).

|              | DROP | ANLI-A3 | BoolQ | WikiQA |
|--------------|------|---------|-------|--------|
| Few-Shot-COT | 57.6 | 52.3    | 67.7  | 65.2   |
| MiniLM           | 63.0 | 53.7 | 70.2 | 67.0 |
| Instructor-base  | 64.2 | 53.2 | 70.2 | 66.9 |
| Random           | 57.5 | 52.8 | 69.7 | 66.3 |
| + MPNet          | 64.7 | 53.3 | 70.4 | 67.1 |
| + LLM-Retrieval  | **65.7** | **55.2** | **71.5** | **68.0** |

Table 3: The comparison of retrieval methods.

## 4.3 Analyses

**Multiple-Decoding Performance** In this section, we evaluate MoT under self-consistency strategy (Wang et al., 2022b) which decodes multiple times and uses majority-voting to get the final answer. We compare MoT with baselines across varying sampling times and temperatures on OBQA, BoolQ and WikiQA, and the results are shown in Table 2. We see that MoT consistently outperforms Zero-Shot-CoT and Few-Shot-CoT across different decoding temperatures and sampling times, which indicates the generality and stability of MoT. We notice that the improvements slightly diminish when using more sampling times. This is similar to the phenomenon in human beings: the more carefully we think about a question, the less our previous preparation matters.

**The Effect of LLM-retrieval** To evaluate the effect of LLM-retrieval for MoT, we conduct experiments with varying retrieval methods on DROP, ANLI-A3, BoolQ and WikiQA, shown in Table 3. Besides the SBERT ("all-mpnet-base-v2", abbreviated as MPNet) (Reimers and Gurevych, 2019) used in MoT, we further compare two other semantic embedders, SBERT ("all-MiniLM-L6-v2", abbreviated as MiniLM) (Reimers and Gurevych, 2019) and Instructor-base (Su et al., 2022b) which is trained by 330 diverse tasks and supports various scenarios. We observe that using only MPNet for memory retrieval also brings significant improvements over Few-Shot-CoT, which shows MoT's usability under the limited LLM-API budget. After using the LLM to retrieve memory, the performance gets further improvements, which directly demonstrates the effectiveness of LLM-retrieval. Additionally, we see that LLM-retrieval outperforms all compared semantic embedders, which shows that the LLM can better capture the complicated reasoning logic than semantic embeddings.

**The Effect of Filtering** To evaluate the effect of memory filtering in MoT, we plot the performance

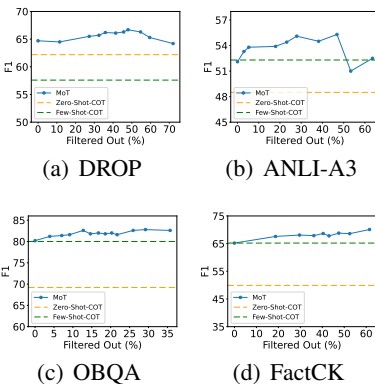

(a) DROP   (b) ANLI-A3

(c) OBQA   (d) FactCK

Figure 5: The effect of filtering. The far left and far right of the x-axis correspond to no filtering, and the strictest filtering, i.e., only the memory that all paths lead to the same answer can be retained.

curve over different filtering thresholds on DROP, ANLI-A3, OBQA and WikiQA. Specifically, we tune the filtering threshold of answer-entropy uniformly and observe the corresponding performance. The results are shown in Figure 5. We find that the MoT without filtering significantly degrades and slightly underperforms Few-shot-COT on some datasets, e.g., OBQA and FactCK, which indicates that the incorrect memory can deteriorate the LLM's reasoning and thus our filtering strategy is necessary. Meanwhile, most filtering thresholds consistently lead to improvements over baselines, which demonstrates that the improvements of MoT exhibit insensitivity to the hyper-parameter of filtering thresholds in general.

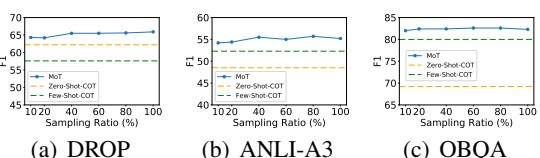

(a) DROP   (b) ANLI-A3   (c) OBQA

Figure 6: Limited Memory-Size Performance.

**Limited Memory-Size Performance** In real-world scenarios, the number of unlabeled examples or the size of available external memory space may be limited, and these can both lead to the limited memory-size. In this section, we evaluate MoT under different memory sizes. Specifically, we conduct experiments on the randomly sampled subsets with different proportions and plot the corresponding performance curve in Figure 6. We observe that MoT can consistently lead to performance improvements. Even under 10 percent of the original memory pool, MoT can still outperform Zero-Shot-CoT and Few-Shot-CoT. These show the usability of MoT when unlabeled examples or available ex-

| | DROP | ANLI-A1 | ANLI-A3 | OBQA |
|---|---|---|---|---|
| Few-Shot-COT | 57.6 | 59.7 | 52.3 | 80.0 |
| +MoT | 65.7 | 65.6 | 55.2 | 82.3 |
| Zero-Shot-COT | 62.2 | 61.9 | 48.5 | 69.2 |
| +MoT | 66.6 | 65.9 | 54.0 | 81.5 |
| Plan-and-Solve | 60.5 | 62.6 | 52.1 | 71.4 |
| +MoT | 67.6 | 66.3 | 56.9 | 85.6 |

Table 4: Comparison of various CoT methods.

ternal memory space are limited.

**Transferability across Different COT Methods**
In this section, we evaluate the performance of
MoT on two additional COT methods: Zero-
Shot-COT (Kojima et al., 2022) and Plan-and-
Solve (Wang et al., 2023). Compared with Zero-
Shot-COT which uses "Let's think step by step" to
elicit LLM's reasoning, Plan-and-Solve uses a spe-
cialized prompt to let the LLM first devise a plan
to divide the entire task into sub-tasks and then
solve them based on the plan, and thus it can ac-
complish more complicated reasoning (Wang et al.,
2023). For these two CoT methods, we use them
to generate the pool of memory-of-thoughts at pre-
thinking stage, respectively. At the test stage, we
retrieve thoughts from the corresponding memory
pool, concatenate them with the test question, and
then use the corresponding prompt, e.g., "Let's
think step by step" for Zero-Shot-COT, to elicit the
LLM's reasoning. Results on DROP, ANLI-A1,
ANLI-A3 and OBQA are shown in Table 4. We ob-
serve that MoT leads to consistent improvements,
which shows its stability and generality across vari-
ous CoT methods. Moreover, since these two CoT
methods do not rely on manual CoT demonstra-
tions, these results also demonstrate the effective-
ness of MoT when the manual CoT demonstration
is not available. Meanwhile, when using the more
advanced CoT method, Plan-and-Solve, MoT's per-
formance gets further improvements, which shows
its potential in the future where the more powerful
CoT method is proposed.

## 5 Related Work

**Model Augmentation by LLM-generated Data**
In this section, we introduce previous methods that
use the data generated by LLMs for model augmen-
tation. Ye et al. (2022a); Gao et al. (2022); Ye et al.
(2022b) propose ZeroGen, ProGen and ZeroGen$^+$
to use the LLM to generate the dataset to enhance
small models, e.g., LSTM. Fu et al. (2023a); Mag-

ister et al. (2022); Ho et al. (2022) leverage LLM
to generate reasoning paths and teach small LMs
to reason. Wang et al. (2022c) and Honovich et al.
(2022) leverage the LLM to generate instruction
data and improve the instruction-following capa-
bility of the LLM. Schick et al. (2023) propose
ToolFormer, which learns how to use various tools
by self-generated data. Zelikman et al. (2022) and
Huang et al. (2022) leverage the LLM to gener-
ate reasoning paths and improve itself using la-
beled and unlabeled datasets, respectively. Dif-
ferent from these methods that depend on expen-
sive fine-tuning, MoT can make the LLM self-
improve with memory-of-thoughts and does not
depend on parameter updates and is compatible
with API-accessing LLM. Recently, Zhang et al.
(2022b); Shao et al. (2023) automatically generate
COT demonstrations by the LLM itself. Li et al.
(2022) leverage the LLM to generate the knowl-
edge base and improve its ability of open-domain
QA. These methods can be seen as the specialized
case of MoT, with task-level memory selection or
task-specialized memory building.

**Demonstration Retrieval for LLM** In this sec-
tion, we introduce previous demonstration retrieval
methods for ICL, which mainly retrieve relevant
input/output pairs, from an annotated dataset, for
the LM to predict the test example. Liu et al.
(2022) propose to leverage a dense semantic em-
bedder to retrieve relevant examples to improve
ICL. Agrawal et al. (2022) leverage BM25 to re-
trieve examples for machine translation's ICL. Das
et al. (2021) and Hu et al. (2022) design specialized
target similarities to train demonstration retrievers
on ICL of knowledge-based question answering
and dialogue state tracking respectively. Rubin
et al. (2022); Shi et al. (2022b) use the LM's feed-
back to train the demonstration retriever for se-
mantic parsing. Lyu et al. (2022) retrieve relevant
examples with random labels and propose heuristic
methods to reduce the negative effect of false la-
bels. Recently, Li et al. (2023) propose UDR, a uni-
fied demonstration retriever for various NLP tasks,
which is trained by the unified LM-feedback on
about 40 annotated datasets. While most of these
methods depend on high-quality annotated datasets
and only explore in-context learning without ra-
tionales, MoT can make the LLM self-improve
without annotated datasets and parameter updates,
and to the best of our knowledge, we are the first to
explore demonstration retrieval in the challenging

and complicated reasoning scenarios and demonstrate MoT's effectiveness.

## 6 Conclusion

In this paper, we propose MoT, a framework that let the LLM self-improve via Memory-of-Thought, without annotated datasets and parameter updates. Experimental results show that MoT can help ChatGPT significantly improve its abilities in arithmetic reasoning, commonsense reasoning, factual reasoning and natural language inference. Further analyses show that each component contributes critically to the improvements and MoT can lead to consistent improvements across various CoT methods and LLMs.

## Limitations

MoT mainly has the following limitations:

- Although we propose the answer-entropy to filter out uncertain thoughts, the remaining thoughts can still contain certain mistakes. We will explore more methods of false thought filtering (Lin et al., 2023) in the future.

- In this paper, we employ a simple strategy to utilize the relevant memory, i.e., concatenate it with the test input $q_{test}$ and thus help the LLM answer $q_{test}$. We will explore more strategies to utilize the retrieved memory, e.g., retrieving the memory to verify the current reasoning path for $q_{test}$.

- On the one hand, in this paper, we make the first step to let the LLM self-improve based on the memory mechanism. The conducted experiments are still in a safe setting, i.e., a specific unlabeled dataset, and the LLM cannot access the internet and control external tools. Hence we think our method and experiment are still safe enough, which will not cause serious impact and unrecoverable consequences on society. On the other hand, large language models have shown various kinds of bias (Bender et al., 2021). Since we let the LLM generate thoughts/memory to help itself, the LLM might suffer from the generated biased content. We see LLM debias as an important future research topic.

## Acknowledgements

This work was supported by the National Natural Science Foundation of China (No. 62236004 and No. 62022027).

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

## A The Example of LLM-Retrieval

We show the LLM-retrieval example in Table 6.

## B Dataset Details

**Overview** We conduct experiments on ten datasets, including four task families:

- **Arithmetic reasoning:** AQuA (Ling et al., 2017): A multi-choice dataset of arithmetic questions covering various topics and difficulty levels, and DROP (Dua et al., 2019): A reading comprehension dataset that needs discrete reasoning;

- **Natural Language Inference:** Adversarial NLI subsets (Nie et al., 2020), including ANLI-A1, ANLI-A2 and ANLI-A3, which cover varying difficulty levels respectively;

- **Commonsense Reasoning:** OBQA (OpenBookQA) (Mihaylov et al., 2018): Commonsense-related questions which require the facts and their applications to novel situations, and ComV (Commonsense Validation) (Wang et al., 2019): A dataset that requires for identifying the sentence that does not make sense from two sentences of similar wording;

- **Factual Reasoning:** BoolQ (Clark et al., 2019), FactCK (Fact Checker) (Srivastava et al., 2022): A dataset that tests the ability to evaluate the authenticity of factual claims covering Wikipedia, COVID-19 and Politics. WikiQA (Srivastava et al., 2022): question answering fron randomly-sampled Wikidata fact triples.

**Split, Evaluation Metric and Statistics** For AQuA, DROP, ANLI-A1, ANLI-A2, ANLI-A3, ComV and OBQA, we use their official test set for evaluation. For BoolQ, we follow Wang et al. (2022a) to use the validation set for evaluation, since its test set is not publicly available. For FactCK and WikiQA, we manually split them into a train/test split, and use the questions of the training set as unlabeled dataset, since there is not split version of them released. Limited by the budget, for the DROP dataset, we only use the half of its unlabeled dataset (the questions of training set) for the LLM to pre-think. For the classification or multi-choice datasets, we use the accuracy as evaluation metric. For the abstractive QA dataset

including DROP and WikiQA, we use the F1-score as evaluation metric. For DROP, since its one test example has multiple annoted answer, we follow its original paper (Dua et al., 2019) to take a max over all annotated answers. Limited by budget, for those evaluation datasets that are larger than 1000, we randomly sample a subset of 1000 examples for evaluation. We list the overall dataset satistiscs, the size of memory after filtering and evaluation metrics in Table 7.

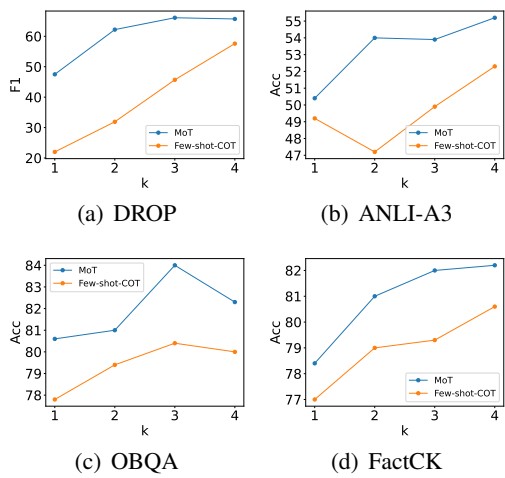

(a) DROP  (b) ANLI-A3

(c) OBQA  (d) FactCK

Figure 7: The impact of demonstration quantity.

## C Performance on Different LLMs

We conduct experiments on Text-Davinci-002 and Text-Davinci-003 (Chen et al., 2023; Ouyang et al., 2022) to evaluate MoT's generality across different LLMs. We show the results in Table 5. We observe that MoT consistently outperforms baselines on these two LLMs, which shows the effectiveness of MoT does not rely on one specific LLM and it can bring further improvements in the future where the more strong LLM is proposed.

## D The Impact of Demonstration Quantity

We compare MoT and Few-Shot-CoT under varying numbers of demonstrations and the results are shown in Figure 7. We see that MoT consistently outperforms Few-Shot-CoT across varying amounts of demonstrations, which shows the stability of MoT. Additionally, the results show that the demonstrations in retrieved memory are more helpful and informative than manual demonstrations in Few-Shot-CoT: specifically, with 1 or 2 demonstrations, MoT can outperform Few-Shot-CoT with 4 demonstrations on OBQA and DROP.

| Method | ANLI-A3 | OBQA | BoolQ | FactCK |
|---|---|---|---|---|
| *Text-Davinci-002* | | | | |
| Zero-Shot-CoT | 34.2 | 47.2 | 40.8 | 52.2 |
| Few-Shot-CoT | 46.9 | 75.4 | 58.8 | 83.6 |
| MoT | 49.1 | 80.0 | 63.9 | 86.2 |
| *Text-Davinci-003* | | | | |
| Zero-Shot-CoT | 46.9 | 64.2 | 62.8 | 47.2 |
| Few-Shot-CoT | 45.3 | 81.6 | 67.7 | 84.0 |
| MoT | 49.3 | 84.6 | 71.0 | 87.0 |

Table 5: Performance comparison on various LLMs.

## E    Implementation Details

We use the public OpenAI language model of "gpt-3.5-turbo-0301" unless otherwise specified and the experiments (Appendix C) on "text-davinci-002/003" show consistent improvements. Due to the limitation of LLM-API budget, we heuristically set the hyper-parameters in the pre-thinking stage, including the generation temperature $T$, the number of decoded reasoning paths $n$. For the filtering threshold $\tau$ and the number of memory clusters $l$, we conduct exploratory experiments on OBQA and find that $\tau = \{0.2, 0.3, 0.4\}$ and $l = \{3, 4, 5\}$ lead to similar performance. Thus we set $\tau = 0.3$ and $l = 4$, respectively. For pre-thinking, we use the temperature $T = 1.2$ to encourage more diverse reasoning paths, and use $n = 16$ reasoning path sampling times, unless otherwise specified. For memory recall, we use SBERT ("all-mpnet-base-v2") (Reimers and Gurevych, 2019) for semantic filtering. Limited by the LLM's max input length, we fix the the number of each cluster's memory candidates as 10 for each dataset. In the test stage, for the stability of results, we use greedy decoding to generate the output, unless otherwise specified. For simplicity, we separately run MoT on each dataset and regard cross-dataset memory recall as future work. Baselines' points are from our implementation, and share the same templates, answer parsing and evaluation as MoT.

For AQuA, OpenBookQA, BoolQ, DROP, ANLI-A1, ANLI-A2 and ANLI-A3, We use the same few-shot CoT examples as those in Wei et al. (2022c), Zhou et al. (2022), Wang et al. (2022b) and Wang et al. (2022a), respectively. For the left datasets that have no publicly released manual CoT demonstrations, we randomly select questions from the training set and use ChatGPT to generate reasoning paths and get their few-shot CoT examples. We list the used Few-Shot-CoT examples in Table 8, 9, 10, 12, 11, 13, 14 and 15.

## F    Ethics Statement

In this paper we make the first step to let the LLM self-improve based on the memory mechanism. The conducted experiments are still in a safe setting, i.e., a specific unlabeled dataset, and the LLM cannot access the internet and control external tools. Hence we think our method and experiment are still safe enough, which will not cause serious impact and unrecoverable consequences on society.

I will provide you with a target question and 10 reference questions. I need you to choose a reference question from "Reference Questions", whose question, train of thought or answer would be most helpful for you to answer the target question. Please note that the following reference QA pairs are presented in a random order without any prioritization.

Target Question:
Machine A puts out a yo-yo every 6 minutes. Machine B puts out a yo-yo every 9 minutes. After how many minutes will they have produced 10 yo-yos? Answer Choices: (A) 24 minutes (B) 32 minutes (C) 36 minutes (D) 64 minutes (E) 72 minutes

Reference Questions:
1.
Q: Two machines, Y and Z, work at constant rates producing identical items. Machine Y produces 5 items in the same time Machine Z produces 2 items. If machine Y takes 9 minutes to produce a batch of items, how many minutes does it take for machine Z to produce the same number of items? Answer Choices: (A) 6 (B) 9 (C) 9 1/2 (D) 22.5 (E) 13 1/2
2.
Q: Two machines, Y and Z, work at constant rates producing identical items. Machine Y produces 30 items in the same time Machine Z produces 38 items. If machine Y takes 19 minutes to produce a batch of items, how many minutes does it take for machine Z to produce the same number of items? Answer Choices: (A) 6 (B) 9 (C) 9 1/2 (D) 15 (E) 13 1/2
3.
Q: Two machines, Y and Z, work at constant rates producing identical items. Machine Y produces 30 items in the same time Machine Z produces 24 items. If machine Y takes 36 minutes to produce a batch of items, how many minutes does it take for machine Z to produce the same number of items? Answer Choices: (A) 60 (B) 90 (C) 9 1/2 (D) 45 (E) 13 1/2
4.
Q: Working alone at its constant rate, machine A produces x boxes in 10 minutes and working alone at its constant rate, machine B produces 2x boxes in 5 minutes. How many minutes does it take machines A and B, working simultaneously at their respective constant rates, to produce 10x boxes? Answer Choices: (A) 13 minutes (B) 14 minutes (C) 15 minutes (D) 16 minutes (E) 20 minutes
5.
Q: Two machines, Y and Z, work at constant rates producing identical items. Machine Y produces 23 items in the same time Machine Z produces 21 items. If machine Y takes 21 minutes to produce a batch of items, how many minutes does it take for machine Z to produce the same number of items? Answer Choices: (A) 6 (B) 9 (C) 9 1/2 (D) 12 (E) 23
6.
Q: Machines X and Y produce bottles at their respective constant rates. Machine X produces k bottles in 6 hours and machine Y produces k bottles in 12 hours. How many hours does it take machines X and Y , working simultaneously , to produce 12k bottles? Answer Choices: (A) 8 (B) 12 (C) 15 (D) 48 (E) 24
7.
Q: Machines X and Y produce bottles at their respective constant rates. Machine X produces k bottles in 6 hours and machine Y produces k bottles in 3 hours. How many hours does it take machines X and Y , working simultaneously , to produce 12k bottles? Answer Choices: (A) 4 (B) 8 (C) 12 (D) 18 (E) 4
8.
Q: Machines X and Y produce bottles at their respective constant rates. Machine X produces k bottles in 4 hours and machine Y produces k bottles in 5 hours. How many hours does it take machines X and Y , working simultaneously , to produce 10k bottles? Answer Choices: (A) 8 2/3 (B) 12 5/3 (C) 15 (D) 18 (E) 22 2/9
9.
Q: Working alone at its constant rate, machine A produces x boxes in 10 minutes and working alone at its constant rate, machine B produces 2x boxes in 5 minutes. How many minutes does it take machines A and B, working simultaneously at their respective constant rates, to produce 6x boxes? Answer Choices: (A) 3 minutes (B) 4 minutes (C) 5 minutes (D) 6 minutes (E) 12 minutes
10.
Q: Machine A can make 350 widgets in 1 hour, and machine B can make 250 widgets in 1 hour. If both machines work together, how much time will it take them to make a total of 900 widgets? Answer Choices: (A) 1 hour and 20 minutes (B) 1 hour and 24 minutes (C) 1 hour and 30 minutes (D) 1 hour and 36 minutes (E) 1 hour and 40 minutes

Which one of the above reference questions is the most helpful question for you to answer the target question? You must choose exactly one reference question to you answer the target question. Your response must end in this format: "The most helpful question is question [index].". For example, if question 5 is your answer, you must end in "The most helpful question is question 5."

The most helpful question is question 10.

Table 6: The example of LLM-Retrieval.

| Task Family | Task | Task Format | Unlabeled Questions | Test Questions | Memory Size | Metric |
|---|---|---|---|---|---|---|
| *Arithmetic Reasoning* | AQuA | Multi Choice | 97467 | 254 | 19334 | Accuracy |
| | DROP | Abstractive QA | 42777 | 1000 | 17066 | F1 |
| *NLI* | ANLI-A1 | Classification | 16946 | 1000 | 9721 | Accuracy |
| | ANLI-A2 | Classification | 45460 | 1000 | 25444 | Accuracy |
| | ANLI-A3 | Classification | 100459 | 1000 | 53402 | Accuracy |
| *Commonsense Reasoning* | ComV | Multi Choice | 10000 | 1000 | 9003 | Accuracy |
| | OBQA | Multi Choice | 4957 | 1000 | 3559 | Accuracy |
| *Factual Reasoning* | BoolQ | Classification | 9427 | 1000 | 6422 | Accuracy |
| | FactCK | Classification | 6654 | 500 | 4782 | Accuracy |
| | WikiQA | Abstractive QA | 19442 | 1000 | 9041 | F1 |

Table 7: The statistics of used datasets and their evaluation metrics.

---

**Q:** John found that the average of 15 numbers is 40. If 10 is added to each number then the mean of the numbers is? Answer Choices: (A) 50 (B) 45 (C) 65 (D) 78 (E) 64

**A:** If 10 is added to each number, then the mean of the numbers also increases by 10. So the new mean would be 50. The answer is (A).

**Q:** If a / b = 3/4 and 8a + 5b = 22,then find the value of a. Answer Choices: (A) 1/2 (B) 3/2 (C) 5/2 (D) 4/2 (E) 7/2

**A:** If a / b = 3/4, then b = 4a / 3. So 8a + 5(4a / 3) = 22. This simplifies to 8a + 20a / 3 = 22, which means 44a / 3 = 22. So a is equal to 3/2. The answer is (B).

**Q:** A person is traveling at 20 km/hr and reached his destiny in 2.5 hr then find the distance? Answer Choices: (A) 53 km (B) 55 km (C) 52 km (D) 60 km (E) 50 km

**A:** The distance that the person traveled would have been 20 km/hr * 2.5 hrs = 50 km. The answer is (E).

**Q:** How many keystrokes are needed to type the numbers from 1 to 500? Answer Choices: (A) 1156 (B) 1392 (C) 1480 (D) 1562 (E) 1788

**A:** There are 9 one-digit numbers from 1 to 9. There are 90 two-digit numbers from 10 to 99. There are 401 three-digit numbers from 100 to 500. 9 + 90(2) + 401(3) = 1392. The answer is (B).

Table 8: Few-shot CoT prompts for AQuA, from Wang et al. (2022b).

**Q:** The Seahawks played the San Francisco 49ers. In the first quarter, the Hawks RB Julius Jones got a 27-yard TD run, along with DT Craig Terrill returning a fumble 9 yards for a touchdown. In the third quarter, the 49ers almost rallied as RB H. J. Torres made a 12-yard TD pass to Lucas Nelly, along with Mare kicking a 32-yard field goal. In the final quarter, Julius Jones got another 11-yard TD. How many yards do the shortest touchdown run and the longest touchdown pass combine for?

**A:** All the touchdown runs are: a 27-yard touchdown run, a 9-yard touchdown run, a 11-yard touchdown run. The smallest number among 27, 9, 11 is 9. So the shortest touchdown run was 9 yards. All the touchdown passes are: a 12-yard touchdown pass. So the longest touchdown pass was 12 yards. So the shortest touchdown run and the longest touchdown pass combine for 9 + 12 = 21 yards. So the answer is 21 yards.

**Q:** In the county, the population was spread out with 23.50% under the age of 18, 8.70% from 18 to 24, 29.70% from 25 to 44, 24.70% from 45 to 64, and 13.30% who were 65 years of age or older. How many more percent are under the age of 18 compared to the 18 to 24 group?

**A:** According to the passage, 23.5% are under the age of 18, and 8.7% are from ages 18 to 24. 23.5% - 8.7% = 14.8%. So the answer is 14.8.

**Q:** Since the 1970s, U.S. governments have negotiated managed-trade agreements, such as the North American Free Trade Agreement in the 1990s, the Dominican Republic-Central America Free Trade Agreement in 2006, and a number of bilateral agreements. In Europe, six countries formed the European Coal and Steel Community in 1951 which became the European Economic Community in 1958. Two core objectives of the EEC were the development of a common market, subsequently renamed the single market, and establishing a customs union between its member states. How many years did the European Coal and Steel Community exist?

**A:** According to the passage, the European Coal and Steel Community was established in 1951 and became the EEC in 1958. 1958 - 1951 = 7. So the answer is 7.

**Q:** The Vikings flew to Bank of America Stadium to face the Carolina Panthers. After a scoreless first quarter, Carolina got on the board with quarterback Matt Moore finding fullback Brad Hoover on a 1-yard TD pass. After yet another scoreless quarter, Carolina sealed the game as Matt Moore completed a 42-yard touchdown pass to wide receiver Steve Smith. How many scoreless quarters were there?

**A:** The first and third quarters were the scoreless quarters. So there are 2 scoreless quarters. So the answer is 2.

Table 9: Few-shot CoT examples for DROP, from Zhou et al. (2022).

Premise:
"Conceptually cream skimming has two basic dimensions - product and geography."
Based on this premise, can we conclude the hypothesis "Product and geography are what make cream skimming work." is true?
OPTIONS:
- yes
- no
- it is not possible to tell

**A:** Based on "cream skimming has two basic dimensions" we can't infer that these two dimensions are what make cream skimming work. The answer is it is not possible to tell.

Premise:
"One of our member will carry out your instructions minutely."
Based on this premise, can we conclude the hypothesis "A member of my team will execute your orders with immense precision." is true?
OPTIONS:
- yes
- no
- it is not possible to tell

**A:** "one of" means the same as "a member of", "carry out" means the same as "execute", and "minutely" means the same as "immense precision". The answer is yes.

Premise:
"Fun for adults and children."
Based on this premise, can we conclude the hypothesis "Fun for only children." is true?
OPTIONS:
- yes
- no
- it is not possible to tell

**A:** "adults and children" contradicts "only children". The answer is no.

Premise:
"He turned and smiled at Vrenna."
Based on this premise, can we conclude the hypothesis "He smiled at Vrenna who was walking slowly behind him with her mother." is true?
OPTIONS:
- yes
- no
- it is not possible to tell

**A:** the premise does not say anything about "Vrenna was walking". The answer is it is not possible to tell.

Table 10: Few-shot CoT prompts for NLI tasks, three subsets of ANLI from Wang et al. (2022a).

**Q:** Poison causes harm to which of the following? (A) a Tree (B) a robot (C) a house (D) a car

**A:** Poison will harm living things, only a tree is a living thing. The answer is (A).

**Q:** As you look deeper into a Marbel you can see (A) the future (B) minut defects (C) colors (D) the other side

**A:** Marbel is not transparent, so you can not see the other side. Marbel does not necessarily have multiple colors. You will see minut defects. The answer is (B).

**Q:** When food is reduced in the stomach (A) the mind needs time to digest (B) take a second to digest what I said (C) nutrients are being deconstructed (D) reader's digest is a body of works

**A:** The food is being deconstructed in the stomach during digestion. The answer is (C).

**Q:** The sun is responsible for (A) puppies learning new tricks (B) children growing up and getting old (C) flowers wilting in a vase (D) plants sprouting, blooming and wilting

**A:** The sun can affect the growing of living things, like plants. The answer is (D).

Table 11: Few-shot CoT examples for OBQA, from Wang et al. (2022a).

**Q:** Which one of the following statements is against common sense? (A) Roses buds eat caterpillars (B) The caterpillar eats the rose bud

**A:** Statement (A) is against common sense as it goes against the natural food chain and the known behavior of roses. Roses are plants and cannot eat or consume other organisms, including caterpillars. The answer is A.

**Q:** Which one of the following statements is against common sense? (A) He threw his house into the trash bin (B) He threw his food waste into the trash

**A:** It is not physically possible to throw a house into a trash bin. Statement (A) goes against the laws of physics and is therefore illogical. The answer is A.

**Q:** Which one of the following statements is against common sense? (A) Because his car was damaged, he received RMB 1000 from electricity company (B) Because his car was damaged, he received RMB 1000 from insurance company

**A:** It does not make logical sense for an electricity company to compensate someone for car damage. It is more reasonable for an insurance company to provide compensation for car damage. The answer is A.

**Q:** Which one of the following statements is against common sense? (A) Because his car was damaged, he received RMB 1000 from electricity company (B) Because his car was damaged, he received RMB 1000 from insurance company

**A:** It does not make logical sense for an electricity company to compensate someone for car damage. It is more reasonable for an insurance company to provide compensation for car damage. The answer is A.

Table 12: Few-shot CoT examples for ComV.

**Q:** does system of a down have 2 singers?

**A:** System of a Down currently consists of Serj Tankian, Daron Malakian, Shavo Odadjian and John Dolmayan. Serj and Daron do vocals, so the band does have two singers. The answer is yes.

**Q:** do iran and afghanistan speak the same language?

**A:** Iran and Afghanistan both speak the Indo-European language Persian. The answer is yes.

**Q:** is a cello and a bass the same thing?

**A:** The cello is played sitting down with the instrument between the knees, whereas the double bass is played standing or sitting on a stool. The answer is no.

**Q:** can you use oyster card at epsom station?

**A:** Epsom railway station serves the town of Epsom in Surrey and is not in the London Oyster card zone. The answer is no.

Table 13: Few-shot CoT examples for BoolQ, from Wang et al. (2022a).

**Q:** On June 2017, the following claim was made: David Lloyd George lost every bid to become prime minister. Was this claim true or false?

**A:** David Lloyd George served as the Prime Minister of the United Kingdom from 1916 to 1922. He also served as the Chancellor of the Exchequer and the Minister of Munitions before becoming Prime Minister. Therefore, the claim that he lost every bid to become Prime Minister is false. The answer is false.

**Q:** On June 2017, the following claim was made: In 1966, George Harrison got married for the first time. Was this claim true or false?

**A:** George Harrison married his first wife, model Pattie Boyd, on January 21, 1966. The answer is true.

**Q:** On June 2017, the following claim was made: Woodrow Wilson did not live during World War I. Was this claim true or false?

**A:** Woodrow Wilson was the President of the United States during World War I, serving from 1913 to 1921. The answer is false.

**Q:** On April 17 2008, the following claim was made: Hillary Clinton has taken over $800,000 from lobbyists. Was this claim true or false?

**A:** According to OpenSecrets.org, a nonpartisan research group that tracks money in politics, Hillary Clinton received over $800,000 in campaign contributions from lobbyists during her 2008 presidential campaign. The answer is true.

Table 14: Few-shot CoT examples for Fact Checker.

**Q:** On June 2017, the following claim was made: David Lloyd George lost every bid to become prime minister. Was this claim true or false?

**A:** David Lloyd George served as the Prime Minister of the United Kingdom from 1916 to 1922. He also served as the Chancellor of the Exchequer and the Minister of Munitions before becoming Prime Minister. Therefore, the claim that he lost every bid to become Prime Minister is false. The answer is false.

**Q:** The native language of Aaron Swartz is?

**A:** Aaron Swartz was born in Chicago, Illinois, United States. Therefore, his native language is most likely English, as it is the primary language spoken in the United States. The answer is English.

**Q:** The religion of Prajadhipok is?

**A:** Prajadhipok was a Buddhist, as Buddhism is the predominant religion in Thailand, where he was the last absolute monarch before the country became a constitutional monarchy. The answer is Buddhism.

**Q:** The country of Valletta is?

**A:** Valletta is the capital city of Malta, which is a small island nation located in the Mediterranean Sea. The answer is Malta.

**Q:** The sport played by Garry Kasparov is?

**A:** Garry Kasparov is a former world chess champion, therefore the sport played by him is chess. The answer is chess.

Table 15: Few-shot CoT examples for WikiQA.