# OpenReview forum: "MoT: Memory-of-Thought Enables ChatGPT to Self-Improve"
_EMNLP/2023/Conference — EMNLP 2023 Main_

### Official Review · Reviewer_E9wd · 2023-08-01

**Soundness:** 4

**Excitement:**

3: Ambivalent: It has merits (e.g., it reports state-of-the-art results, the idea is nice), but there are key weaknesses (e.g., it describes incremental work), and it can significantly benefit from another round of revision. However, I won't object to accepting it if my co-reviewers champion it.

**Missing References:**

None. Related work and its connections have been discussed in this paper

**Paper Topic And Main Contributions:**

This paper proposed a framework called Memory Of Thought (MoT), which uses self-thinking and memory candidates to help large language models (LLMs) improve their performance. The framework consists of two stages to improve the performance of LLMs in various reasoning tasks by means of pre-thinking and external memory. The main contribution is to propose a method based on self-thinking and memory, which can significantly improve the ability of ChatGPT in multiple reasoning tasks.

The MoT framework proposed in this paper is innovative and practical for solving the improvement problem of LLMs. By means of self-reflection and memory, LLMs can be self-improved without external resources. Experiments demonstrate that MoT can effectively improve the performance of LLMs in multiple key tasks, and individual components play a key role in achieving these improvements. These results have important reference value for the research community in further improving and optimizing LLMs.

**Questions For The Authors:**

As the simply implementation for this memory of thought method:

I have the following questions:

Q1: It is mentioned in the paper that answer entropy is used to filter uncertain thinking, so how to define and calculate answer entropy? Does this approach work consistently across different types of tasks and datasets?

Q2: In the current experimental setup, the paper mentions that LLM cannot access the Internet and control external tools. How does this constraint affect the effectiveness of the MoT framework? Are there plans to conduct experiments in more open settings to further validate the performance of the framework?

Q3: Can we expect more comparative experiments on MoT frameworks and other CoT methods in future research? How to ensure the fairness and reliability of these comparative experiments?

Q4: For the bias problem of large language models, how to ensure that no bias is introduced in the thinking and memory process generated by LLM? Has the generated thinking content been analyzed for bias? How to address and mitigate the problems that arise from potentially biased generated content?

Q5: For unknowable and invisible out-of-domain data sets, is there any experiment to illustrate the generalization of the proposed method? That would be a nice move.

**Reasons To Accept:**

1. Innovativeness: The MoT framework proposed in this paper is innovative in solving the problem of LLMs improvement, and realizes self-improvement without external resources by utilizing self-thinking and memory.

2. Experimental results support: The author has proved through experiments that the MoT framework has a significant improvement effect on ChatGPT, and can improve the abilities of arithmetic reasoning, common sense reasoning, factual reasoning and natural language reasoning.

3. Scalability: The framework can achieve consistent improvement on different CoT methods and LLMs, and has certain versatility and scalability.

**Reasons To Reject:**

1. The proposed memory module is relatively simple to implement. It is not yet known whether the rate of errors in the memory will affect the final reasoning result. This paper does not conduct further analysis about this section.

2. In the implementation process of the article, there are many hyperparameters that need to be adjusted (as mentioned in paper, such as the decoding path, pre-defied threshod \tau, the number of memory volumes and the number of routes for memory reasoning). The paper has not conducted further ablation experiments, and it is not clear how each parameter affects the reasoning.

3. There are many similar ways to use the retrieval function to improve generation, the essence of which lies in the "quality of prompt information". When the article considers the memory module, the performance of the module, including the capacity of the memory block and the way of sending a block of storage (natural language, key Values are equal). Does it affect the final reasoning result?

4. The efficiency of performing more operations in the retrieved memory, or what the utilization rate of the memory is? It is a wise choice to use the visual reasoning path to improve the interpretability of the model reasoning.

I will be glad to revise my rating scores when I see the authors try to take my advice.

**Reproducibility:**

4: Could mostly reproduce the results, but there may be some variation because of sample variance or minor variations in their interpretation of the protocol or method.

**Reviewer Confidence:**

2: Willing to defend my evaluation, but it is fairly likely that I missed some details, didn't understand some central points, or can't be sure about the novelty of the work.

**Typos Grammar Style And Presentation Improvements:**

The article is well organized, but there are still some minor typographical errors that need to be corrected, such as:

1. Line 540 across Please change to uppercase Acrosss

2. Please use the past tense to express the conclusion

---

> ### Author Rebuttal · Authors · 2023-08-29
>
> Thank you for your detailed review and great suggestions. We have organized our responses according to the "Reasons to Reject" (denoted by **R**) and "Questions" (denoted by **Q**).
>
> **R1**: The Effect of Errors in the Memory
>
> The errors in the memory will degrade the performance of MoT. **As shown in Table 1**, MoT (with gold) is the MoT with memory where incorrect memory is filtered out by the gold labels. MoT without gold labels significantly underperforms MoT (with gold), and this shows that **filtering out incorrect memory helps MoT conduct better reasoning**. To further demonstrate this, we conducted the following **supplementary experiments** across multiple tasks, demonstrating a direct correlation between incorrect memory and decreased accuracy.
>
> **Supplementary Experiments**
>
> We conduct a fine-grained experiment to analyze the effect of incorrect memory to the performance on AQuA, ANLI-A3 and ComV over three task families. Specifically, we divide the test examples into multiple groups according to the accuracy of their recalled memory (the answer accuracy of recalled thought), and separately calculate the accuracy on these divided examples. The following is the result.
>
> In the following results, we keep the total number of memory thoughts as in the paper, 4.
> |       | The number of correct memory |  4   |  3   |  2   |  1   |
> |:-----:|:---------------------------:|:----:|:----:|:----:|:----:|
> | AQuA  |          Accuracy           | **57.8** | 54.2 | 51.0 | ///  |
> | ANLI-A3|         Accuracy           | **58.0** | 54.2 | 50.6 | 50.0 |
> | ComV  |          Accuracy           | **96.6** | 93.8 | ///  | ///  |
>
>
> Note: Accuracy calculations for groups with insufficient samples, such as (2,1 in ComV), are omitted.
>
> From the results above, we see that **with more incorrect memory, it is slightly easier for the LLM to make mistakes**, which directly shows the effectiveness of memory filtering. We will add these experiments and conduct further case studies of incorrect memory in the next version.
>
>
> **R2**: The **Effect of each Hyper-parameter**, e.g.,  the Decoding path, Pre-defined Threshold $\tau$, the Number of Memory Volumes and the Number of Routes for Memory Reasoning
>
> We agree with you that it is important to analyze the effect of hyper-parameters, and **we have conducted analyses of these hyper-parameters in the submitted version** as follows:
>
> **Decoding Path**: We analyze the performance of MoT under different decoding paths in Section 4.3 "Multiple-Decoding Performance" (Line 469)
>
> **Pre-defined Threshold $\tau$**: We compare the performance of MoT under different filtering thresholds in section 4.3 "The Effect of Filtering" (Line 508)
>
> **The Number of Memory Volumes**: We analyze the performance of MoT when the memory size is limited in section 4.3 "Limited Memory-Size Performance" (Line 525)
>
> **The Number of Routes for Memory Reasoning**: We analyze the effect of recalled memory quantity in Appendix D  (Line 1229) and we will move it to the main body of the paper in the next version.
>
> We will further conduct hyper-parameter analyses in the next version.
>
>
> **R3**: The Effect of Capacity of the Memory Block and the Way of Sending a Block of Storage
>
> As mentioned in R2, we have conducted the analysis of the capacity of the memory block (entire memory capacity: section 4.3 "Limited Memory-Size Performance" (Line 525), recalled memory capacity: Appendix D  (Line 1229)). The results show that **MoT consistently outperforms baselines over the different recalled memory capacities**. And MoT with larger recalled memory capacity can get better performance.
>
> As for the way of sending memory, we considered different memory orders. We found that **different orders of recalled memory in the prompt lead to similar results**, as in the previous work [1]. We will add this result and conduct further analyses about the memory module in the next version.
>
> If we misunderstand the meaning of your mentioned "the effect of capacity of the memory block and the way of sending a block of storage", please tell us.
>
> **R4**: The Utilization Rate of the Recalled Memory, Visualization.
>
> Since we conduct experiments on the OpenAi API (ChatGPT, Text-Davinci-002,003), we cannot get access to the model's internal representation e.g., attention score, and it is hard for us to visualize the memory visualization. We will **add the experiments on LLaMA models, and further analyze and visualize the effect of memory** in the next version. We will conduct case studies to directly illustrate the effect of memory in the next version, too.
>
> **Q1**: The Calculation of Answer-Entropy and Whether It Consistently Works across Different Types of Tasks and Datasets?
>
> We introduced the definition and calculation of answer-entropy in **Section 3.1.2 and Equations (3), (4), (5)**. Specifically, we count the frequency of each answer in the multiple-decoding paths as their probability, and use the probabilities of these answers to calculate the entropy of the answer's distribution. The larger the entropy is, the more uncertain the LLM is about its answer.
>
> In section 4.3 "The Effect of Filtering" (Line 508), Figure 5, we show **MoT without filtering significantly underperforms MoT with filtering**, across four task families. Specifically, the performance of MoT without filtering degrades to the Few-Shot CoT. These directly show the effect of filtering, which depends on the proposed answer-entropy.
>
> **Q2**: ""In the current experimental setup, the paper mentions that LLM cannot access the Internet and control external tools. How does this constraint affect the effectiveness of the MoT framework?""
>
> This constraint forces the MoT to depend on its own ability to self-improve, and our experiments show that it can self-improve without external resources. Accessing external environments like the internet or tools can help the LLM self-think better, thereby generating better memory to help itself think and improve its various abilities. Since the small-scale tool-using is relatively safer than accessing the internet for the AI, we will add the experiments where the LLM can use tools to self-improve in the next version.
> Meanwhile, ensuring the security of AI systems is important, and we regard making LLM self-improve both effectively and safely in an open environment as important future work.
>
> **Q3**: Experiments on Other CoT Methods and How to Ensure the Fairness and Reliability of These Comparative Experiments
>
> We analyzed the performance of MoT on other two CoT methods (Zero-shot CoT and Plan-and-Solve) in **section 4.3 "Transferability across Different COT Methods" (Line 540)**. These experiments demonstrate that **MoT is compatible with various CoT methods and can consistently make LLM self-improve using various CoT methods**, which shows the generality and extensibility of MoT. We will add the experiments on more CoT methods in the next version.
>
> How to conduct fair and reliable comparative experiments?
>
> We think the most important thing is to control variables. Specifically, when comparing MoT and baselines, we should ensure the other factor is the same, e.g., evaluation metric, the inference LLM, the CoT method, the test dataset and so on. Since our proposed MoT can be compatible with various CoT methods, when applying one CoT method on MoT, we should incorporate the used CoT method into baselines as shown in Table 4 (Line 553).
>
> We list the **implementations (including hyper-parameters) and used prompting templates in Appendix A and E**. The dataset details are in Appendix B. As mentioned in the submitted paper, we will **release the code and the generated CoT reasoning paths to facilitate future research**. We hope that MoT can inspire researchers of the important design choices about making the LLM self-improve with memory mechanisms and pave the way for further improvements.
>
> **Q4**: How to Mitigate the Problems that Arise from Potentially Biased Generated Content?
>
> We can add an extra post-processing procedure to filter out those biased generated memory. For example, we can make the LLM itself **a bias classifier by prompting, and recognize and filter out those biased generated content**, or we can use the LLM itself to **refine those generated memory and debias them**. [2]
>
> **Q5**: Generalization on Out-of-Domain Dataset
>
> For this setting, we test MoT on ANLI-A3 (which is the most difficult level in ANLI datasets) but using the memory from ANLI-A1 (which is the easiest level in ANLI datasets) and the following is the result:
>
> ANLI-A3
> |           Method            | Accuracy |
> |:---------------------------:|:--------:|
> |       Zero-Shot-CoT         |   48.5   |
> |       Few-Shot-CoT          |   52.3   |
> | MoT with ANLI-A3 memory     |   55.2   |
> | MoT with ANLI-A1 memory     |   54.4   |
>
> We see that MoT still outperforms baselines when it uses the out-of-domain memory, and this shows that the **MoT can leverage the memory from easy reasoning scenarios to enhance its reasoning ability for difficult scenarios**. This result shows its potential utility in challenging real-world scenarios. We will further add the OOD generation experiments in the next version.
>
> **Typos and Grammars**: The typo on line 540 and the tense in the conclusion
>
> Thanks for pointing out these issues. We will fix the typo and use the past tense to express the conclusion in the next version.
>
> Thank you again for your valuable review. If we have misunderstood any aspect of your review, or if you have additional questions, please let us know.
>
>
>
>
> [1] Unified Demonstration Retriever for In-Context Learning
>
> [2] Self-Refine: Iterative Refinement with Self-Feedback

---

### Official Review · Reviewer_qXWz · 2023-08-03

**Typos Grammar Style And Presentation Improvements:** 1. The figures (flg.4, 5, 6) are too …
**Soundness:** 4

**Excitement:**

4: Strong: This paper deepens the understanding of some phenomenon or lowers the barriers to an existing research direction.

**Paper Topic And Main Contributions:**

This paper proposes a framework, Memory-of-Thought (MoT), designed to enhance the self-improvement capabilities of Large Language Models (LLMs) without the need for expensive fine-tuning or annotated datasets.

Inspired by the human ability to improve through metacognition and memory mechanisms, the MoT framework allows the LLM to self-improve, preserving its general capabilities. This approach takes its cues from how memory aids human decision-making, reasoning, and judgment.

The MoT method uses a two-stage process: firstly, it "pre-thinks" on an unlabeled dataset, saving high-confidence thoughts as an external memory; secondly, during a test phase, the LLM uses this memory to aid its reasoning and responses to given questions.

The researchers report that their experiments show MoT can significantly enhance ChatGPT's abilities in arithmetic reasoning, commonsense reasoning, factual reasoning, and natural language inference without the need for parameter updates or annotated datasets. They also note that each component of MoT contributes critically to the improvements and that the framework consistently enhances various CoT methods and LLMs.

The key contributions of this paper include:

1. The creation of MoT framework allowing an LLM to improve its reasoning abilities using memory mechanisms, without parameter updates and annotated datasets.
2. Comprehensive experimentation showing that the MoT framework can lead to significant improvements in various areas for ChatGPT without requiring annotated datasets or parameter updates.
3. The promise to make the code and generated CoT reasoning paths available to the wider research community, potentially inspiring future research and improvement efforts.


**Questions For The Authors:**





1. Most of the experiments are carried out on ChatGPT. It would be also interesting to test your approach on open-source LLMs, such as LLaMA, Llama-2, etc.


**Reasons To Accept:**



1. The creation of MoT framework allowing an LLM to improve its reasoning abilities using memory mechanisms, without parameter updates and annotated datasets.
2. Comprehensive experimentation showing that the MoT framework can lead to significant improvements in various areas for ChatGPT without requiring annotated datasets or parameter updates.
3. The promise to make the code and generated CoT reasoning paths available to the wider research community, potentially inspiring future research and improvement efforts.




**Reasons To Reject:**

I have no major reasons to reject this paper. As a minor comment, the authors should have uploaded their code/resources to reviewers but there is none. As the authors say they will open-source their code/resources, I will not blame on this too much.

**Reproducibility:**

2: Would be hard pressed to reproduce the results. The contribution depends on data that are simply not available outside the author's institution or consortium; not enough details are provided.

**Reviewer Confidence:**

3: Pretty sure, but there's a chance I missed something. Although I have a good feel for this area in general, I did not carefully check the paper's details, e.g., the math, experimental design, or novelty.

---

> ### Author Rebuttal · Authors · 2023-08-29
>
> Thank you for your comprehensive review and insightful suggestions. Below is our response to the comments and questions.
>
> **Q1**: Regarding experiments on LLaMA models
>
> We agree with you that whether the method is effective on the open-source model is important. In this version, we conduct experiments on ChatGPT and Text-Davinci-002,003 (please see Appendix C), and we will add the experiments on LLaMAs in the next version.
>
> **Q2**: Figures 4,5,6 are too small and the typos on line 309
>
> Thanks for pointing out these issues, and we will fix them in the next version.
>
>
>
> **Q3**: Low Reproducibility
>
> We have noted your concern about the reproducibility of our study.
>
> We have provided the following details in the submitted paper:
>
> **Used Prompting Templates**: Appendix A
>
> **Implementations  (including hyper-parameters)**: Appendix E
>
> **Dataset Details**: Appendix B, the used datasets are all publicly available.
>
> We will release the code and the generated CoT reasoning paths after review, to facilitate future research.
>
> However, we understand that more information might be needed to fully address this issue. Could you kindly specify which implementation details or hyper-parameters you found to be insufficient for reproducibility?
>
>
> Thank you again for your valuable review. If we have misunderstood any aspect of your review, or if you have additional questions, please let us know.

---

### Official Review · Reviewer_v4Dz · 2023-08-11

**Soundness:** 4

**Excitement:**

4: Strong: This paper deepens the understanding of some phenomenon or lowers the barriers to an existing research direction.

**Paper Topic And Main Contributions:**

In this paper, the authors introduce a framework called MoT, which enables the Large Language Model (LLM) to self-enhance using the Memory-of-Thought technique. Initially, numerous examples are generated from unlabeled datasets, which are then treated as external memory for use during inference. In the testing phase, instead of relying on semantic embedders, a large language model is employed to recall the top-k most relevant memory examples, which are then prepended to the query in order to get the final answer. Experimental findings demonstrate that MoT significantly enhances ChatGPT's performance in areas such as arithmetic reasoning, commonsense reasoning, factual reasoning, and natural language inference.

**Reasons To Accept:**

* The paper is meticulously structured and articulated, making it exceptionally easy to follow and understand.
* The author propose a nice method, which does not required labeled datasets but can achieve better performance than the original few-shot in context learning.
* The paper carries out various experiments for the MOT framework on four different task families, providing strong evidence of its effectiveness and practical applicability.

**Reasons To Reject:**

* What most confuse me is that I am not sure how much novalty the method MOT has. There are some self-prompting methods have been proposed early (such as [1]). This paper considers the self-prompting example as the memory and propose some new filtering and selecting method to recall the most helpful one, which has limited novelty for me. Could you discuss more about what is the different between your work and the previous self-prompting work.

[1] Self-Prompting Large Language Models for Zero-Shot Open-Domain QA - Li et al.

**Reproducibility:**

5: Could easily reproduce the results.

**Reviewer Confidence:**

4: Quite sure. I tried to check the important points carefully. It's unlikely, though conceivable, that I missed something that should affect my ratings.

---

> ### Author Rebuttal · Authors · 2023-08-29
>
> Thank you for your comprehensive review and insightful suggestions. Below is our response to the comments and questions.
>
>
> **Q1**: What distinguishes our proposed method, MoT, from existing self-prompting work, "Self-Prompting Large Language Models for Zero-Shot Open-Domain QA - Li et al"?
>
> At a high level, MoT and self-prompting are guided by distinct motivations, conceptual perspectives, and design methodologies. MoT leverages the **memory mechanism** to enhance various capabilities of Large Language Models (LLMs) like ChatGPT over challenging reasoning scenarios, without annotated datasets and parameter updates. In contrast, self-prompting primarily focuses on utilizing LLMs to generate wiki-QA pairs for one single language understanding task, open-domain QA.
>
> When Self-prompting depends on **a specialized pipeline** (passage generation, entity recognition, question generation, QA explanation) to generate Wikipedia QA pairs for open domain QA, MoT is proposed to make LLM, e.g., ChatGPT, self-improve based on the memory mechanism. To the best of our knowledge, MoT is **the first framework** to make the LLM self-improve over various abilities based on the memory mechanism.
>
> Specifically, MoT consists of these components:
>
> **1. Memory Generation:** This enables the LLM to **self-think** to build the external memory pool, which is both extensible and compatible with **various CoT** methods.
>
> **2. Memory Filtering:** This filters out the generated incorrect memory by **the LLM's own uncertainty**.
>
> **3. Memory Recalling:** The **LLM retrieves helpful memory by itself** (while self-prompting uses the typical dual-encoder).
>
> As a result, the MoT can make the LLM self-improve various abilities over challenging reasoning scenarios: arithmetic reasoning, commonsense reasoning, factual reasoning, and natural language inference, while the self-prompting method is specialized for one natural language understanding task, open domain QA.
> We hope that MoT can inspire researchers of the important design choices about making the LLM self-improve with memory mechanisms and pave the way for further improvements.
>
> Our further analysis also shows that MoT is effective and compatible with different CoT methods or LLMs (while self-prompting depends on one specialized data generation pipeline), and its performance improves as the CoT methods or the used LLMs improve, which shows MoT's generality and extensibility, and its potential in the future where more powerful CoT methods are proposed.
>
> We discussed the difference between MoT and self-prompting in the section "Related Work" (Line 596~600) and we will discuss more about this difference to clarify the novelty of MoT in the next version.
>
> Thank you again for your valuable review. If we have misunderstood any aspect of your review, or if you have additional questions, please let us know.

---

### Meta-Review · Area_Chair_MEXY · 2023-09-25

**Recommendation:** 4

**Metareview:**

improvement of large language models (LLMs) without fine-tuning or annotated data. The framework uses a two-stage process: first, it generates and stores high-confidence examples from an unlabeled dataset as external memory; second, it recalls and uses the most relevant memory examples to assist the LLM in answering questions. The paper shows that MoT can significantly improve ChatGPT’s performance on various reasoning tasks, such as arithmetic, commonsense, factual, and natural language inference. The paper also demonstrates the importance of each component of MoT and its generalizability to different LLMs and methods.

---

### Decision · Program_Chairs · 2023-10-07

**Decision:**

Accept-Main

**Comment:**

improvement of large language models (LLMs) without fine-tuning or annotated data. The framework uses a two-stage process: first, it generates and stores high-confidence examples from an unlabeled dataset as external memory; second, it recalls and uses the most relevant memory examples to assist the LLM in answering questions. The paper shows that MoT can significantly improve ChatGPT’s performance on various reasoning tasks, such as arithmetic, commonsense, factual, and natural language inference. The paper also demonstrates the importance of each component of MoT and its generalizability to different LLMs and methods.